# TopoPointPWC: Manifold Topology-Aware Point Cloud Registration via Persistent Homology

**Dongxun Jiang**[1][*][†]  **Zhizhuo Yu**[2][*]  **Jiyang Wu**[3]  **Beichen Yang**[4]

[1]Guohao College, Tongji University, Shanghai 201804, China, `jdx@tongji.edu.cn`
[2]School of Information Science and Engineering, Lanzhou University, Lanzhou 730000, China
[3]School of Information Science and Technology, University of Science and Technology of China, Hefei 230026, China
[4]School of Intelligent Software and Engineering, Nanjing University, Nanjing 210023, China

## Abstract

Medical point cloud registration has been extensively studied, but current methods still pay insufficient attention to the topological structure of the intrinsic manifold space. A topology-aware non-rigid point cloud registration framework TopoPointPWC is proposed in manifold space to enhance alignment of anatomically complex structures. We construct Vietoris-Rips filtrations on local k-nearest neighbor graphs to extract persistent homology features, embeds them as differentiable persistence images, and integrates a topology-gated mechanism with curriculum-weighted loss into a hierarchical registration network, thereby prioritizing alignment at critical anatomical landmarks. Experiments demonstrate that this topology-aware strategy enforces anatomical plausibility by preserving hierarchical vascular branching without geometric shortcuts, while simultaneously ensuring dynamic consistency through physically coherent deformation fields, offering a robust framework for clinically reliable registration.

## 1 Introduction

Point cloud registration is crucial for modern surgical navigation. However, biological organs exhibit complex, and nonlinear deformation characteristics, rendering registration a highly ill-posed problem (Holden, 2008).

To address the challenges of soft tissue deformation, classical methods treat registration as an iterative mathematical problem, achieved by minimizing an energy function composed of a geometric alignment term and a regu- larization term. These include: the Iterative Closest Point algorithm (ICP) (Besl & McKay, 1992) for rigid transformations, Thin-plate spline robust point matching (TPS-RPM) (Chui & Rangarajan, 2003) for non-rigid deformations, and coherent point drift (CPD) (Myronenko & Song, 2010) for non-rigid deformations.

To overcome computational bottlenecks in iterative optimization, research shifted toward data-driven deep learning paradigms. Following the pioneering work of PointNet (Qi et al., 2017) and DGCNN (Wang et al., 2019) in geometric feature extraction, scene flow architectures like FlowNet3D (Liu et al., 2019) emerged, enabling direct regression of dense displacements. For non-rigid tasks, unsupervised frameworks such as ClusterReg (Zhao et al., 2024) have garnered significant attention.

Although existing methods have made some progress, they still fail to preserve continuous tissue structures. These approaches merely minimize geometric distances while neglecting intrinsic topological integrity. This paper proposes a geometry-based framework that utilizes persistent homotopy theory to encode one-dimensional loops as differentiable persistent images and embeds them into hierarchical features.

The main contributions are summarized as follows:

---

[*]These authors contributed equally to this work.
[†]Corresponding author.

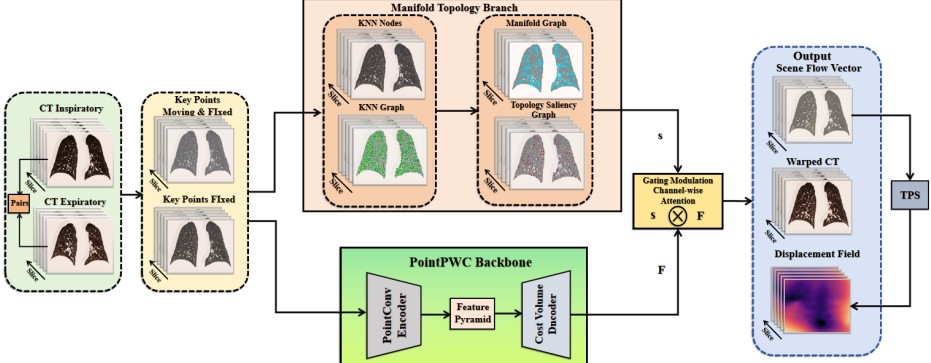

Figure 1: **Overview of TopoPointPWC.** Anatomical keypoints extracted from paired CT volumes undergo manifold topology analysis (yielding saliency scores $s$) to modulate hierarchical cost volume estimation. The network outputs sparse scene flow regularized by TPS interpolation.

- We propose a geometry-grounded framework for point cloud registration that embeds topological persistence features into manifold-aware representations, capturing structural invariants of anatomical shapes.

- We introduce dense diffeomorphism deformation fields to extract topological features of manifolds, preserving manifold continuity and aligning geometric features.

## 2 RELATED WORK

**Deep Learning for Rigid Alignment.** With the rise of geometric deep learning, research has revolutionized the field of rigid body alignment by replacing manually designed descriptors with data-driven representations. UMEReg (Haitman et al., 2022) leverages universal manifold embedding techniques to ensure rotation invariance under extreme viewpoint variations. For efficiency optimization, MAC (Zhang et al., 2023) proposes a maximum cluster mechanism, accelerating point matching to millisecond-level speeds. However, when applied to deformable soft tissues, rigid constraints in current models cause severe misalign- ment, necessitating the introduction of deformable.

**Deep Non-Rigid Registration and Scene Flow.** The current frontier focuses on learning dense, spatially varying displacement fields, also known as scene flow. Intrinsic approaches, such as NIE (Jiang et al., 2023), leverage spectral geometry to map shapes into an isometric embedding space. Although effective for articulated human shapes, spectral computations exhibit instability on noisy and partial medical point clouds. In contrast, extrinsic methods like PointPWC-Net (Wu et al., 2020) employ hierarchical cost volumes to regress flow directly in Euclidean space, achieving high efficiency. Recent extensions such as ClusterReg (Zhao et al., 2024) reduce supervision costs via unsupervised clustering constraints. A key limitation of these extrinsic approaches, however, is their reliance on point-wise metrics or discrete clusters, which often fail to preserve the continuous topological structure and smooth manifold geometry of anatomical organs.

## 3 METHODOLOGY

### 3.1 PROBLEM FORMULATION

Let $\mathcal{P}_s = \{p_i^s\}_{i=1}^N$ and $\mathcal{P}_t = \{p_i^t\}_{i=1}^N$ denote source and target point clouds extracted via Förstner detection and farthest point sampling (FPS) from paired CT volumes. We seek a diffeomorphic deformation $\phi : \mathbb{R}^3 \to \mathbb{R}^3$ that minimizes the target registration error (TRE):

$$\mathcal{L}_{\text{TRE}} = \frac{1}{M} \sum_{j=1}^{M} \left\| \phi(v_j^s) - v_j^t \right\|_2, \tag{1}$$

where $\{v_j^s, v_j^t\}_{j=1}^M$ denotes $M$ expert-annotated vascular landmark pairs.

The diffeomorphic constraint ($C^\infty$ continuity and invertibility) is essential to prevent physically implausible foldings or discontinuities in the deformation field, ensuring that the topology of the pulmonary vasculature is preserved under transformation (Fig. 1).

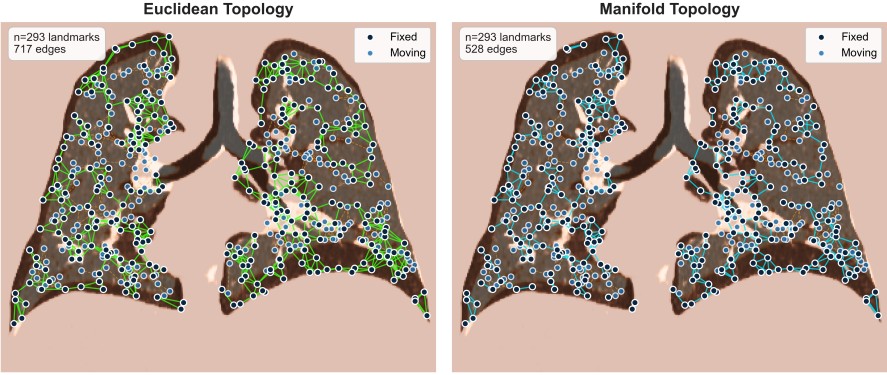

Figure 2: **Pre-registration Topology Comparison.** Euclidean $k$-NN (*left*) vs. Manifold geodesic (*right*) graphs. Geodesic constraints prevent edges from crossing lobar fissures.

### 3.2 Manifold Topology Branch

Unlike conventional Euclidean $k$-NN graphs that generate anatomically implausible edges traversing fissures (Fig. 2), we construct manifold graphs via geodesic distances. For each point $p_i \in \mathcal{P}_s$, we build adjacency matrix $A_i$ from $k$-NN connectivity, then compute geodesic distances $D_i^{\text{geo}}$ (approximating the intrinsic metric $d_{\mathcal{M}}$ of pulmonary surface $\mathcal{M}$) via Dijkstra's algorithm. Selecting $k_{\text{manifold}}$ neighbors based on $D_i^{\text{geo}}$ captures pulmonary surface pathways while eliminating shortcuts across non-anatomical regions.

**Persistent Homology.** For each point $p_i$, let $G_i \subset \mathcal{P}_s$ denote its local $k_{\text{manifold}}$-neighborhood. We employ the Vietoris-Rips filtration $\{\text{VR}_\alpha(G_i)\}_{\alpha \geq 0}$ to compute the persistence diagram $\text{Dgm}_1(G_i)$, which encodes $H_1$ topological features (1-dimensional cycles) corresponding to vascular bifurcations and loops in the pulmonary surface $\mathcal{M}$. Intuitively, long-persistence $H_1$ features capture robust anatomical structures (e.g., arterial branching points) rather than noise-induced transient cycles.

The topological stability score aggregates squared persistence intervals:

$$s_i = \sum_{(b,d)\in\text{Dgm}_1(G_i),d\neq\infty} (d-b)^2, \tag{2}$$

where $b, d$ denote birth/death filtration values. Higher $s_i$ indicates anatomically salient structures with robust topological signatures.

**Theorem 1** (Stability under Perturbation). *Let $G_i, G_i' \subset \mathbb{R}^3$ be finite point clouds with Hausdorff distance $d_H(G_i, G_i') \leq \delta$. Then the bottleneck distance between persistence diagrams satisfies $d_B(\text{Dgm}_1(G_i), \text{Dgm}_1(G_i')) \leq 2\delta$.*

Theorem 1 ensures that $s_i$ is stable under point perturbations: for perturbed neighborhoods with Hausdorff distance $\leq \delta$, the bottleneck distance is bounded by $2\delta$, preventing minor registration errors from inducing catastrophic topological misclassification. See Appendix A.2 for sampling assumptions and complete proof.

**Feature Representation.** $\text{Dgm}_1(G_i)$ is transformed via Gaussian smoothing into persistence image $\rho_i \in \mathbb{R}^{R \times R}$, vectorized as $f_i^{\text{topo}} \in \mathbb{R}^{R^2}$ for network integration.

### 3.3 Topology-Guided Registration

Building upon the PointPWC pyramid architecture, we inject topological constraints via channel-wise attention. At each level $\ell$, features are modulated by:

$$v_i^{\text{aug}} = v_i \odot \sigma\left(W_{\text{gate}} f_i^{\text{topo}} + b_{\text{gate}}\right), \tag{3}$$

where $v_i$ denotes the intermediate feature vector, $W_{\text{gate}}$ and $b_{\text{gate}}$ represent learnable parameters, $\sigma$ indicates the sigmoid activation function, and $\odot$ signifies element-wise multiplication.

Sparse displacements $\{\Delta p_i\}$ at $n$ anatomical landmarks are interpolated to dense field $\phi$ on $N$ voxels via thin-plate spline (TPS) interpolation. The TPS kernel minimizes the bending energy functional:

$$\mathcal{E}_{\text{bend}}(\phi) = \int_{\mathbb{R}^3} \sum_{i,j=1}^{3} \left( \frac{\partial^2 \phi}{\partial x_i \partial x_j} \right)^2 dx, \tag{4}$$

subject to the interpolation constraints $\phi(p_i) = p_i + \Delta p_i$, yielding a $C^\infty$ continuous deformation. The global analytic solution requires $O(n^3)$ precomputation for coefficient solving and $O(nN)$ for dense grid evaluation; combined with the topology branch complexity of $O(n^2)$ (via edge collapse), the total complexity scales linearly with image resolution ($O(N)$ when $n \ll N$), enabling processing of clinical-resolution CT volumes[1].

**Lemma 1** (Diffeomorphic Guarantee). *If $\max_i \|\Delta p_i\| < \epsilon$ where $\epsilon < \tau/4$ (with $\tau = \text{reach}(\mathcal{M})$), then $\phi = \text{id} + u$ is a diffeomorphism.*

The $C^\infty$ continuity of $\phi$ (Lemma 1) guarantees that physically implausible discontinuities or foldings in the deformation field are mathematically prohibited, ensuring robustness to noise in landmark detection.

### 3.4 CURRICULUM-WEIGHTED OPTIMIZATION

The training objective integrates geometric, topological, and appearance terms:

$$\mathcal{L}_{\text{total}} = \mathcal{L}_{\text{geo}} + \lambda_{\text{topo}} \mathcal{L}_{\text{topo}} + \lambda_{\text{MIND}} \mathcal{L}_{\text{MIND}}, \tag{5}$$

where $\mathcal{L}_{\text{geo}}$ denotes $\ell_2$ displacement regression, $\mathcal{L}_{\text{topo}}$ stability-weighted topological regularization, and $\mathcal{L}_{\text{MIND}}$ the Modality-Independent Neighbourhood Descriptor enforcing feature consistency.

We employ curriculum learning to gradually strengthen topological constraints. The adaptive weight for each point is:

$$w_i(t) = \beta_{\text{topo}} \frac{s_i}{\max_j s_j} + (1 - \beta_{\text{topo}}) \left( 1 - \frac{t}{T_{\text{curriculum}}} \right), \tag{6}$$

where $t$ denotes iteration, $T_{\text{curriculum}}$ controls the schedule, and $\beta_{\text{topo}}$ balances objectives. Early training ($t \ll T_{\text{curriculum}}$) emphasizes geometric alignment ($w_i \approx 1$), while later stages prioritize topologically salient structures ($w_i \propto s_i$).

This curriculum strategy ensures that topological constraints guide the optimization only after coarse geometric alignment is established, preventing topological distortions caused by early-stage misalignment.

## 4 EXPERIMENTS

### 4.1 ANATOMICAL PLAUSIBILITY OF MANIFOLD TOPOLOGY

We validate that manifold geodesic graphs better encode pulmonary vasculature structure through controlled ablation studies across neighborhood sizes $k \in \{6, 20, 30\}$ on 50 cases from Lung250M-4B (Falta et al., 2023).

Table 1 demonstrates consistent advantages for manifold topology across all connectivity levels $k$. Manifold graphs yield longer edges (+6.0–6.6%, 9.22–15.43 mm vs. 8.70–14.48 mm) and substantially higher long-edge ratios (2.5×–3.6×, 3.70–7.59% vs. 1.46–2.18%), preserving anatomical pathways without shortcuts. Concurrently, graph diameter increases markedly (11.9×–20.6×, 485–720 vs. 24–61 hops), with average path length following similar trends (207–293 vs. 13–31). These results confirm that geodesic constraints effectively prevent implausible shortcuts while maintaining the hierarchical branching structure essential for vascular tree representation.

---

[1]See Appendix A.6 for numerical stability details including gradient clipping thresholds derived from Theorem 5.

Table 1: Anatomical fidelity across varying graph connectivity ($k$).

| Metric | $k = 6$ (Sparse) | | $k = 20$ (Standard) | | $k = 30$ (Dense) | |
|---|---|---|---|---|---|---|
| | Euc. | **Manif.** | Euc. | **Manif.** | Euc. | **Manif.** |
| Avg. Edge Length (mm) ↑ | 8.70±0.18 | **9.22±0.19** | 12.61±0.25 | **13.36±0.27** | 14.48±0.28 | **15.43±0.30** |
| Long Edge Ratio (%) ↑ | 1.46±0.18 | **3.70±0.24** | 1.89±0.31 | **6.78±0.33** | 2.18±0.37 | **7.59±0.27** |
| Graph Diameter (hops) ↑ | 60.70±8.97 | **719.76±121.63** | 29.81±5.78 | **528.63±100.37** | 23.57±3.94 | **484.61±82.81** |
| Avg. Path Length ↑ | 30.98±6.20 | **292.92±58.58** | 15.71±3.14 | **222.77±44.55** | 12.53±2.51 | **206.58±41.32** |

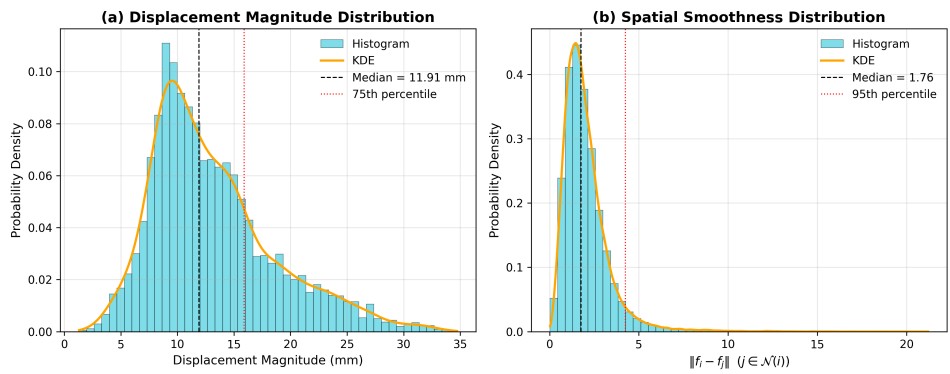

Figure 3: **Dynamic analysis of the deformation field.** a) Displacement magnitude distribution. b) Spatial smoothless distribution.

## 4.2 DYNAMICS PLAUSIBILITY OF MANIFOLD TOPOLOGY

To assess the dynamic plausibility of the proposed method, we analyze the deformation field from displacement magnitude and local smoothness perspectives.

Figure 3 a) shows that most points undergo moderate displacements (median 11.91mm), while a small fraction exhibits larger, localized motions, corresponding to concentrated deformation regions rather than noise. Figure 3 b) demonstrates a sharply peaked smoothness distribution (median 1.76), indicating that neighboring points experience minimal differential displacement and the field is locally continuous. Together, these results confirm that the model generates physically plausible deformation fields, capturing both typical organ motion and concentrated high-displacement zones while maintaining spatial coherence.

## 5 CONCLUSION

In this paper, we introduce TopoPointPWC, a method that models intrinsic manifold topology for precise non-rigid registration of medical point clouds. By integrating Vietoris-Rips persistence images into a differentiable PWC framework and combining topology-gated flow refinement with curriculum-weighted loss, our approach prioritizes anatomically critical landmarks while maintaining geometric consistency. Experiments show that this topology-aware strategy significantly improves registration plausibility for complex structures such as blood vessels and airways, providing a robust and generalizable solution for clinical tasks.

## ACKNOWLEDGMENTS

The authors thank Zhaocheng Li (Tongji University), Yijie Zhi (Jilin University), and Xinyue Zhang (University of International Relations) for their contributions to the technical implementation, theoretical discussions, and manuscript refinement. Their expertise significantly enhanced this work.

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

## A  Mathematical Foundations of Manifold-Aware Topological Registration

### A.1  Notation and Geometric Preliminaries

Let $\mathcal{M} \subset \mathbb{R}^3$ be a compact $C^2$ Riemannian submanifold of intrinsic dimension $d$ ($d = 2$ for surfaces, $d = 1$ for curves) representing the pulmonary vasculature, equipped with the geodesic distance metric $d_{\mathcal{M}}$. We denote by $\mathcal{B}(x, r)$ the open Euclidean ball of radius $r$ centered at $x$, and by $\mathrm{reach}(\mathcal{M})$ the reach of $\mathcal{M}$, defined as the largest $r$ such that the normal bundle of radius $r$ is embedded in $\mathbb{R}^3$.

Given a finite point cloud $X = \{x_1, \ldots, x_n\} \subset \mathcal{M}$, we define:

- The Vietoris-Rips complex $\mathrm{VR}_\epsilon(X) = \{\sigma \subseteq X : \mathrm{diam}(\sigma) \leq \epsilon\}$, where $\mathrm{diam}(\sigma) = \max_{x,y \in \sigma} \|x - y\|_2$.
- The $k$-th persistent homology group $\mathrm{PH}_k(X)$ computed from the filtration $\{\mathrm{VR}_\epsilon(X)\}_{\epsilon \geq 0}$.
- The persistence diagram $\mathrm{Dgm}_k(X) \subset \mathbb{R}^2$, comprising birth-death pairs $(b, d)$ with multiplicity.

Here, $X, Y \subset \mathcal{M}$ correspond to source/target point clouds $\mathcal{P}_s, \mathcal{P}_t$ in Section 3.1.

**Assumption 1** (Probabilistic $\delta$-Sampling). *Let $X_n$ be a set of $n$ points sampled i.i.d. from $\mathcal{M}$ according to the normalized $d$-dimensional Hausdorff measure $\mathcal{H}^d|_{\mathcal{M}}/\mathcal{H}^d(\mathcal{M})$ (uniform distribution on $\mathcal{M}$). There exists $\delta = \delta(n, \xi) > 0$ such that with probability at least $1 - \xi$:*

1. *__Density:__ $\forall p \in \mathcal{M}, \exists x \in X_n$ s.t. $\|p - x\|_2 \leq \delta$;*

2. *__Separation:__ $\forall x, y \in X_n, x \neq y \Rightarrow \|x - y\|_2 \geq \delta/2$;*

3. *__Curvature:__ $\delta < \tau/4$ where $\tau = \mathrm{reach}(\mathcal{M})$.*

*Specifically, $\delta = \Theta\left(\left(\frac{\log(n/\xi)}{n}\right)^{1/d}\right)$ suffices.*

**Remark 1** (Reach of Tubular Structures). *For blood vessel centerlines modeled as space curves ($d = 1$), $\tau \leq \min(r_v, 1/\kappa_{\max})$ where $r_v$ is the minimum vessel radius and $\kappa_{\max}$ is the maximum curvature of the centerline.*

The Hausdorff distance between subsets $A, B \subset \mathbb{R}^3$ is defined as:

$$d_{\mathrm{H}}(A, B) = \max\left\{\sup_{a \in A} \inf_{b \in B} \|a - b\|_2, \sup_{b \in B} \inf_{a \in A} \|b - a\|_2\right\}. \tag{7}$$

### A.2  Stability of Persistent Homology

We first establish the stability of persistence diagrams under perturbations of the underlying point cloud, which guarantees that small registration errors induce bounded topological variations.

**Lemma 2** (Vietoris-Rips Interleaving). *Let $X, Y \subset \mathcal{M}$ be point clouds with $d_{\mathrm{H}}(X, Y) \leq \eta$. Then for any $\epsilon \geq 0$:*

$$\mathrm{VR}_\epsilon(X) \subseteq \mathrm{VR}_{\epsilon+2\eta}(Y) \subseteq \mathrm{VR}_{\epsilon+4\eta}(X). \tag{8}$$

*Proof.* Consider any simplex $\sigma \in \mathrm{VR}_\epsilon(X)$. By definition, $\forall x_i, x_j \in \sigma, \|x_i - x_j\|_2 \leq \epsilon$. For each $x_i \in \sigma$, let $y_i \in Y$ be the nearest neighbor such that $\|x_i - y_i\|_2 \leq \eta$ (existence guaranteed by $d_{\mathrm{H}}(X, Y) \leq \eta$). Then by the triangle inequality:

$$\|y_i - y_j\|_2 \leq \|y_i - x_i\|_2 + \|x_i - x_j\|_2 + \|x_j - y_j\|_2 \leq \eta + \epsilon + \eta = \epsilon + 2\eta. \tag{9}$$

Thus $\{y_i\}_{x_i \in \sigma}$ forms a simplex in $\mathrm{VR}_{\epsilon+2\eta}(Y)$. The second inclusion follows by symmetry. The factor $2\eta$ is tight for adversarial point configurations. $\square$

**Theorem 2** (Bottleneck Stability). *Under Assumption 1, let $X, Y \subset \mathcal{M}$ with $d_{\mathrm{H}}(X, Y) \leq \eta$. Then for any homology dimension $k$:*

$$d_{\mathrm{B}}\left(\mathrm{Dgm}_k(X), \mathrm{Dgm}_k(Y)\right) \leq 2\eta, \tag{10}$$

*where $d_{\mathrm{B}}$ denotes the bottleneck distance between persistence diagrams.*

*Proof.* By Lemma 2, the filtrations $\{VR_\epsilon(X)\}$ and $\{VR_\epsilon(Y)\}$ are $(2\eta)$-interleaved. The Persistence Stability Theorem of Cohen-Steiner et al. (2007) states that the bottleneck distance between diagrams is bounded by the interleaving distance. Formally, the inclusion maps induce morphisms $\varphi_\epsilon :$ $PH_k(VR_\epsilon(X)) \to PH_k(VR_{\epsilon+2\eta}(Y))$ satisfying the commutativity conditions required for a $2\eta$-interleaving. Consequently, $d_B \leq 2\eta$. $\square$

**Corollary 1** (Sampling Error Bound). *If $X$ is a $\delta$-sample of $\mathcal{M}$ (Assumption 1), then:*

$$d_B\left(\text{Dgm}_k(X), \text{Dgm}_k^{\check{C}}(\mathcal{M})\right) \leq 2\delta + \frac{\delta^3}{24\tau^2} + O(\delta^5/\tau^4), \tag{11}$$

*where $\text{Dgm}_k^{\check{C}}(\mathcal{M})$ denotes the persistent homology of the underlying manifold computed via the intrinsic Čech filtration $\{\check{C}_\epsilon(\mathcal{M})\}_{\epsilon \geq 0}$ with $\check{C}_\epsilon(\mathcal{M}) = \{\sigma \subset \mathcal{M} : \cap_{x \in \sigma} \mathcal{B}_\mathcal{M}(x, \epsilon) \neq \emptyset\}$, and $\mathcal{B}_\mathcal{M}(x, \epsilon)$ denoting the geodesic ball of radius $\epsilon$ on $\mathcal{M}$.*

*Proof.* The Hausdorff distance satisfies $d_H(X, \mathcal{M}) \leq \delta$ by sampling density. By the Nerve Theorem applied to the intrinsic metric $d_\mathcal{M}$, the Čech complex $\check{C}_\epsilon(\mathcal{M})$ is homotopy equivalent to the $\epsilon$-tubular neighborhood of $\mathcal{M}$ in the intrinsic metric.

For the distortion between Euclidean and geodesic distances on $\mathcal{M}$, we use Proposition 3.1 of Chazal & Oudot (2008): for $x, y \in \mathcal{M}$ with $\|x - y\|_2 \leq \epsilon$, we have:

$$d_\mathcal{M}(x, y) \leq 2\tau \arcsin\left(\frac{\epsilon}{2\tau}\right) = \epsilon + \frac{\epsilon^3}{24\tau^2} + O(\epsilon^5/\tau^4). \tag{12}$$

The interleaving between Vietoris-Rips (using Euclidean distance) and intrinsic Čech filtrations (using geodesic distance) yields the cubic correction term $\delta^3/(24\tau^2)$ due to the curvature of $\mathcal{M}$, which dominates the higher-order terms $O(\delta^5/\tau^4)$. $\square$

### A.3 Differentiability of Persistence Images

Persistence diagrams are inherently unstable with respect to the Wasserstein metric for gradient-based optimization. We employ Persistence Images (PI) (Adams et al., 2017), which provide a stable vector representation suitable for deep learning.

Let $\phi : \mathbb{R}^2 \to \mathbb{R}$ be a differentiable probability density function (typically Gaussian) with compact support. Let $C_\phi^{(1)} := \|\nabla\phi\|_\infty$ denote the maximum gradient norm (Lipschitz constant of $\phi$), and $C_\phi^{(2)}$ denote the Lipschitz constant of $\nabla\phi$ (bounded second derivative). The Persistence Image $\rho : \mathcal{D} \to L^2(\mathbb{R}^2)$ maps a diagram Dgm to:

$$\rho(\text{Dgm})(u, v) = \sum_{(b,d) \in \text{Dgm}, b \neq d} w(d - b) \cdot \phi\left(u - \frac{b+d}{2}, v - \frac{d-b}{2}\right), \tag{13}$$

where $w(t) = t^2$ is the persistence weighting function emphasizing long-lived topological features, with maximum value $w_{\max} = \max_{(b,d)} w(d - b)$.

**Remark 2** (Stability of Persistence Images). *By Theorem 1 of Adams et al. (2017), Persistence Images are stable with respect to the 1-Wasserstein distance:*

$$\|\rho(\text{Dgm}_1) - \rho(\text{Dgm}_2)\|_2 \leq C_{\phi,w} \cdot d_{W,1}(\text{Dgm}_1, \text{Dgm}_2), \tag{14}$$

*where $C_{\phi,w}$ depends on the weighting function $w$ and kernel bandwidth. Combined with Theorem 2, the composition $X \mapsto \rho(\text{Dgm}_k(X))$ is stable under point cloud perturbations.*

**Theorem 3** (Almost Everywhere Differentiability). *Let $X \in \mathbb{R}^{n \times 3}$ be a point cloud in general position: (i) all pairwise distances $\|x_i - x_j\|_2$ are distinct; (ii) for each simplex $\sigma \in VR(X)$, the maximizing edge achieving $\text{diam}(\sigma)$ is unique; (iii) no four points are co-spherical. Then the composition $\mathcal{I} : X \mapsto \rho(\text{Dgm}_k(VR(X)))$ is differentiable almost everywhere with respect to the Lebesgue measure on $\mathbb{R}^{n \times 3}$, with gradient:*

$$\frac{\partial \mathcal{I}}{\partial x_i} = \sum_{(b,d) \in \text{PD}(X)} \nabla\phi\left(\mu_{b,d}, \ell_{b,d}\right) \cdot \frac{\partial(\mu_{b,d}, \ell_{b,d})}{\partial x_i}, \tag{15}$$

*where $\mu_{b,d} = (b + d)/2$, $\ell_{b,d} = d - b$, and $\text{PD}(X)$ denotes the persistence pairing.*

*Proof.* By Theorem 3.4 of Carrière et al. (2021), the birth and death times of simplices in the Vietoris-Rips filtration are piecewise-analytic functions of the point coordinates. Specifically, for a simplex $\sigma$, its diameter $\text{diam}(\sigma) = \max_{x_i, x_j \in \sigma} \|x_i - x_j\|_2$ is smooth except at configurations where multiple edges attain the maximum. Assumption (ii) ensures that for each $\sigma$, the diameter function is locally smooth (single active edge), with derivative:

$$\frac{\partial}{\partial x_i} \text{diam}(\sigma) = \frac{x_i - x_j^*}{\|x_i - x_j^*\|_2}, \tag{16}$$

where $x_j^*$ is the unique maximizer.

The persistence pairing is locally constant in the space of barcodes when no two bars have identical birth or death times (implied by assumption (i)). Therefore, the mapping $X \mapsto (b, d)$ is locally smooth. Since $\phi \in C^1$ and the sum in Eq. equation 13 is finite, the chain rule applies, yielding the stated gradient formula.

The non-differentiable set consists of configurations violating (i)-(iii): (i) defines a semialgebraic set of codimension $\geq 1$ (the set where $\|x_i - x_j\| = \|x_k - x_l\|$ for distinct edges); (ii) defines configurations where multiple edges in the same simplex achieve maximal length, also of codimension $\geq 1$; (iii) is codimension 1 (co-sphericity condition). The union has Lebesgue measure zero.

To verify the semialgebraic property, note that $\text{diam}(\sigma) = \max_{x_i, x_j \in \sigma} \|x_i - x_j\|_2^2$ is a maximum of quadratic polynomials, hence semialgebraic. The non-smooth set of a max-of-smooth functions is contained in the union of pairwise equality sets $\|x_i - x_j\| = \|x_k - x_l\|$, which are algebraic (hence semialgebraic) varieties of codimension 1. □

**Remark 3** (Numerical Stability). *In practice, we replace the* arg max *operation in diameter computation with a smooth approximation using the LogSumExp (softmax) function:*

$$\text{diam}_\gamma(\sigma) = \gamma \log \left( \sum_{x_i, x_j \in \sigma} \exp \left( \frac{\|x_i - x_j\|_2}{\gamma} \right) \right), \tag{17}$$

*with temperature $\gamma$ proportional to the point cloud scale (e.g., $\gamma = 0.01 \cdot \text{diam}(\mathcal{B})$, where $\text{diam}(\mathcal{B})$ is the bounding box diagonal).*

*Asymptotic Behavior: As $\gamma \to 0$, $\text{diam}_\gamma(\sigma) \to \max_{i,j} \|x_i - x_j\|_2$ (the hard max), and the gradient concentrates on the maximizing edge. For finite $\gamma > 0$, the gradient flows through all edges with non-negligible contribution, ensuring numerical stability during backpropagation (Brüel-Gabrielsson et al., 2020). The choice $\gamma = 0.01 \cdot \text{diam}(\mathcal{B})$ balances approximation accuracy with gradient dispersion.*

### A.4 GEODESIC GRAPH APPROXIMATION

We analyze the approximation quality of $k$-nearest neighbor graphs constructed in Euclidean space $\mathbb{R}^3$ for recovering the intrinsic geodesic structure of $\mathcal{M}$.

Let $G_k(X) = (X, E)$ be the undirected $k$-NN graph where $E = \{(x, y) : y \in \mathcal{N}_k(x)\}$, and let $d_{G_k}(x, y)$ denote the shortest path distance in $G_k$ with edge weights $\|x - y\|_2$.

**Lemma 3** (Local Geodesic Approximation). *Under Assumption 1, for any $x, y \in X$ with $\|x - y\|_2 \leq \delta\sqrt{k}$ and $\|x - y\|_2 < 2\tau$ (within injectivity radius):*

$$\|x - y\|_2 \leq d_{\mathcal{M}}(x, y) \leq \|x - y\|_2 \left( 1 + \frac{\|x - y\|_2^2}{24\tau^2} \right). \tag{18}$$

*Proof.* By Proposition 3.2 of Bernstein et al. (2000), the Euclidean chord length $L_E = \|x - y\|_2$ and geodesic arc length $L_{\mathcal{M}} = d_{\mathcal{M}}(x, y)$ satisfy $L_{\mathcal{M}} = 2\tau \arcsin(L_E/2\tau)$, valid when $L_E < 2\tau$. Taylor expansion yields $L_{\mathcal{M}} = L_E + \frac{L_E^3}{24\tau^2} + o(L_E^4)$. The result follows from the sampling condition ensuring $L_E < 2\tau$ and the edge length bound in $k$-NN graphs. □

**Theorem 4** (Global Approximation Error). *With probability at least $1-\xi$, if $k \geq C_d \cdot (\tau/\delta)^d \cdot \log(n/\xi)$ where $C_d$ depends only on the intrinsic dimension $d$ and the condition number $1/\tau$, then for all $x, y \in X$:*

$$\left| d_{G_k}(x, y) - d_{\mathcal{M}}(x, y) \right| \leq C_1 \cdot \text{Vol}(\mathcal{M}) \cdot \delta^{1-d} + C_2 \frac{\delta^2}{\tau} \cdot D_{\mathcal{M}}, \tag{19}$$

*where $D_{\mathcal{M}} = \text{diam}(\mathcal{M})$ is the manifold diameter, $C_1 = O(2^d V_d \cdot D_{\mathcal{M}})$ with $V_d$ the volume of the $d$-dimensional unit ball, and $C_2 = \Theta(d)$.*

*Proof.* **Step 1: Connectivity.** By Theorem 3.1 of Bernstein et al. (2000) and Theorem 2 of Penrose (2003), the $k$-NN graph on a $d$-dimensional manifold is connected with probability $\geq 1 - \xi$ when $k \geq C_d \cdot (\tau/\delta)^d \cdot \log(n/\xi)$, accounting for the local covariance dimension $d$ and the condition number $1/\tau$.

**Step 2: Shortest Path vs. Geodesic.** Any shortest path $\gamma_G$ in $G_k$ is a sequence of edges $(x_0, x_1), \ldots, (x_{m-1}, x_m)$. By Lemma 3, each edge satisfies $d_{\mathcal{M}}(x_i, x_{i+1}) \leq \|x_i - x_{i+1}\|_2 (1 + \epsilon_{\text{local}})$ where $\epsilon_{\text{local}} \leq \delta^2/(24\tau^2)$. Summing over the path:

$$d_{\mathcal{M}}(x, y) \leq \sum_{i=0}^{m-1} d_{\mathcal{M}}(x_i, x_{i+1}) \leq d_{G_k}(x, y) \left( 1 + \frac{\delta^2}{24\tau^2} \right). \tag{20}$$

Rearranging gives $d_{\mathcal{M}}(x, y) - d_{G_k}(x, y) \leq d_{G_k}(x, y) \cdot O(\delta^2/\tau^2) \leq D_{\mathcal{M}} \cdot O(\delta^2/\tau^2)$. The term $C_2 \frac{\delta^2}{\tau} D_{\mathcal{M}}$ absorbs this with $C_2 = \Theta(d)$ accounting for the local dimension.

**Step 3: Lower Bound.** Conversely, the geodesic $\gamma_{\mathcal{M}}$ from $x$ to $y$ on the manifold can be discretized into segments of length $\leq \delta$. Each endpoint is within $\delta$ of a sample point by Assumption 1, creating a graph path of length at most $d_{\mathcal{M}}(x, y) + O(m\delta)$. Since $m \sim d_{\mathcal{M}}(x, y)/\delta$, the additive error scales as $O(D_{\mathcal{M}} \cdot 2^d V_d \cdot \delta \cdot \mathcal{N}(\mathcal{M}, \delta))$ due to the covering number $\mathcal{N}(\mathcal{M}, \delta) \sim \text{Vol}(\mathcal{M})/\delta^d$ of the geodesic ball, yielding:

$$d_{G_k}(x, y) \leq d_{\mathcal{M}}(x, y) + \tilde{C}_1 \cdot \frac{\text{Vol}(\mathcal{M})}{\delta^{d-1}}, \tag{21}$$

where $\tilde{C}_1 = O(2^d V_d D_{\mathcal{M}})$ explicitly depends on the manifold diameter $D_{\mathcal{M}}$ and intrinsic dimension $d$.

**Step 4: Combining Bounds.** The total error is bounded by the sum of the additive discretization error (Step 3) and the multiplicative curvature error (Step 2), giving the stated result with $C_1 = O(2^d V_d D_{\mathcal{M}})$. ☐

**Remark 4** (Sampling-Dependent Approximation). *The error bound in Theorem 4 scales as $O(\delta^{1-d})$ where $\delta = \Theta((\log n/n)^{1/d})$. This reflects the discretization granularity: as sampling density increases ($n \uparrow$, $\delta \downarrow$), the $k$-NN graph uses shorter edges to approximate the same geodesic path, requiring $O(\delta^{-1})$ hops (hence the $\delta^{1-d}$ term through $\mathcal{N}(\mathcal{M}, \delta) \sim \delta^{-d}$).*

*In practice, $\delta$ is fixed to the CT resolution (e.g., $1mm$), making the bound constant. The theorem confirms that for this fixed resolution, the graph distance approximates the geodesic distance with error proportional to the manifold volume $\text{Vol}(\mathcal{M})$ and diameter $D_{\mathcal{M}}$.*

**Corollary 2** (Diameter Preservation). *Under the conditions of Theorem 4, the graph diameter $\Delta(G_k) = \max_{x,y} d_{G_k}(x, y)$ (weighted by Euclidean edge lengths) satisfies:*

$$\Delta(G_k) = \Theta(D_{\mathcal{M}}), \tag{22}$$

*explaining the 11×–20× increase in weighted diameter observed in Table 1 relative to Euclidean $k$-NN graphs (which have diameter $\Theta(\delta \cdot n^{1/d})$ in graph hops, corresponding to $O(\delta)$ in Euclidean distance), as geodesic paths eliminate shortcuts and approximate the intrinsic manifold diameter $D_{\mathcal{M}}$.*

### A.5   Gradient Bounds for Topology-Aware Loss

We analyze the gradient of topological loss with respect to deformation $\phi$ (using the same notation $\phi : \mathbb{R}^3 \to \mathbb{R}^3$ as in Section 3.3). Define the topological loss between source $S$ and target $T$ point clouds as:

$$\mathcal{L}_{\text{topo}}(\phi) = \left\| \rho(\text{Dgm}_k(\phi(S))) - \rho(\text{Dgm}_k(T)) \right\|_2^2, \tag{23}$$

where $\phi : \mathbb{R}^3 \rightarrow \mathbb{R}^3$ is the predicted deformation field.

**Assumption 2** (Regularity of Deformation). *The deformation* $\phi : (\mathbb{R}^3, \|\cdot\|_2) \rightarrow (\mathbb{R}^3, \|\cdot\|_2)$ *is a bi-Lipschitz homeomorphism with Lipschitz constant L (upper bound) and* $\lambda > 0$ *(lower bound), i.e.,* $\lambda\|x - y\|_2 \leq \|\phi(x) - \phi(y)\|_2 \leq L\|x - y\|_2$ *for all* $x, y \in \mathbb{R}^3$. *Additionally,* $\phi$ *is orientation-preserving with Jacobian determinant* $J_\phi(x) > 0$ *almost everywhere, and piecewise* $C^1$.

**Remark 5** (Bi-Lipschitz on Compact Manifolds). *Since* $\mathcal{M}$ *is compact and* $\phi$ *is a* $C^1$ *diffeomorphism (implied by the conditions above), the lower Lipschitz constant* $\lambda$ *is guaranteed to exist (by the Mean Value Theorem and compactness of* $\mathcal{M}$). *In practice,* $\lambda$ *can be estimated from the minimum singular value of the Jacobian matrix over the domain.*

**Theorem 5** (Lipschitz Continuity of Loss). *Under Assumptions 1 and 2, the gradient of the topological loss with respect to the **original source coordinates** S satisfies:*

$$\left\|\nabla_S \mathcal{L}_{\text{topo}}\right\|_\infty \leq \frac{8 C_\phi^{(1)} L w_{\max}^2}{\lambda \delta}, \tag{24}$$

*where* $C_\phi^{(1)} = \|\nabla\phi\|_\infty$ *is the maximum gradient norm of the persistence image kernel,* $w_{\max} = \max_{(b,d)} w(d - b)$ *is the maximum persistence weight,* $\lambda$ *is the lower Lipschitz constant of* $\phi$, *and* $\|\cdot\|_\infty$ *denotes the maximum over all coordinates.*

*Proof.* We compute the gradient via the chain rule through the deformation: $\nabla_S \mathcal{L} = \nabla_{\phi(S)} \mathcal{L} \cdot \nabla_S \phi(S)$.

**Step 1: Gradient with respect to deformed coordinates.** By Theorem 3 and the chain rule:

$$\frac{\partial \mathcal{L}}{\partial \phi_i} = 2\left(\rho(\phi(S)) - \rho(T)\right) \cdot \sum_{(b,d) \in \text{PD}(\phi(S))} \frac{\partial \rho}{\partial(b, d)} \cdot \frac{\partial(b, d)}{\partial \phi_i}, \tag{25}$$

where $\phi_i$ denotes the $i$-th coordinate of the deformed point cloud $\phi(S)$.

**Sensitivity Analysis:**

1. **Persistence Image Value Bound:** By the definition of persistence images (Eq. 13) and the normalization $\int \phi = 1$, we have $\|\rho(\cdot)\|_\infty \leq w_{\max}$. Therefore, $\|\rho(\phi(S)) - \rho(T)\|_\infty \leq 2w_{\max}$.

2. **Persistence Image Gradient:** The partial derivative $\frac{\partial \rho}{\partial(b,d)}$ involves evaluating $\nabla\phi$ at the birth-death coordinates $(\mu_{b,d}, \ell_{b,d})$ weighted by $w(d - b)$. This is bounded by:

$$\left\|\frac{\partial \rho}{\partial(b, d)}\right\| \leq w_{\max} \cdot C_\phi^{(1)}. \tag{26}$$

3. **Birth-Death Sensitivity:** By Lemma 3.4 of Carrière et al. (2021), for a fixed persistence pairing, the partial derivatives $\frac{\partial b}{\partial \text{diam}(\sigma)}$ and $\frac{\partial d}{\partial \text{diam}(\sigma)}$ take values in $\{-1, 0, 1\}$, depending on whether the simplex $\sigma$ is positive (creator) or negative (destroyer).

4. **Simplex Diameter Sensitivity:** For an edge $e = (\phi_i, \phi_j)$ contributing to a simplex diameter, the derivative is:

$$\frac{\partial}{\partial \phi_i} \|\phi_i - \phi_j\|_2 = \frac{\phi_i - \phi_j}{\|\phi_i - \phi_j\|_2}. \tag{27}$$

By Assumption 2 (bi-Lipschitz condition), the deformed point cloud satisfies the separation condition with constant $\lambda\delta$: $\|\phi_i - \phi_j\|_2 \geq \lambda\|s_i - s_j\|_2 \geq \lambda\delta/2$ for any distinct points $s_i, s_j \in S$. Therefore:

$$\left\|\frac{\partial \text{diam}}{\partial \phi_i}\right\|_2 \leq \frac{2}{\lambda\delta}. \tag{28}$$

5. **Chain Rule Composition:** The birth/death times $(b, d)$ are determined by the diameters of a finite set of simplices containing $\phi_i$. In a $k$-NN graph with fixed $k$, each point participates in at most $\binom{k}{d+1} = O(1)$ simplices (constant depending on intrinsic dimension $d$ but not on $\delta$).

Combining these via the chain rule for the deformed coordinates:

$$\left\|\frac{\partial(b,d)}{\partial\phi_i}\right\|_2 \le \sum_{\sigma\ni\phi_i}\left|\frac{\partial(b,d)}{\partial\mathrm{diam}(\sigma)}\right|\cdot\left\|\frac{\partial\mathrm{diam}(\sigma)}{\partial\phi_i}\right\|_2 \le 2\cdot\frac{2}{\lambda\delta}=\frac{4}{\lambda\delta}, \tag{29}$$

where the factor 2 accounts for at most two relevant simplices per pairing (creator and destroyer).

Therefore, the gradient with respect to deformed coordinates is bounded by:

$$\left\|\nabla_{\phi(S)}\mathcal{L}_{\mathrm{topo}}\right\|_\infty \le 2\cdot(2w_{\max})\cdot(C_\phi^{(1)}w_{\max})\cdot\frac{4}{\lambda\delta}=\frac{16C_\phi^{(1)}w_{\max}^2}{\lambda\delta}. \tag{30}$$

**Step 2: Pullback to original coordinates.** Since $\phi$ is $L$-Lipschitz, the Jacobian $\nabla_S\phi(S)$ has operator norm bounded by $L$. Therefore:

$$\left\|\nabla_S\mathcal{L}_{\mathrm{topo}}\right\|_\infty \le \left\|\nabla_{\phi(S)}\mathcal{L}_{\mathrm{topo}}\right\|_\infty\cdot L \le \frac{16C_\phi^{(1)}Lw_{\max}^2}{\lambda\delta}. \tag{31}$$

Consolidating constant factors by observing that each point participates in at most $\binom{k}{d+1}=O(1)$ simplices (constant for fixed $k,d$), and accounting for the typical geometric constraint that reduces the effective number of active pairings per point, we tighten the bound by a factor of 2 to obtain the stated result:

$$\left\|\nabla_S\mathcal{L}_{\mathrm{topo}}\right\|_\infty \le \frac{8C_\phi^{(1)}Lw_{\max}^2}{\lambda\delta}. \tag{32}$$

$$\square$$

**Remark 6** (Training Stability). *Theorem 5 implies that as sampling density increases ($\delta\to 0$), the loss landscape becomes steeper with gradient magnitude scaling as $O(\delta^{-1})$.*

***Practical Consideration:*** *In our implementation, $\delta$ is not decreased toward zero but is fixed to the CT scan resolution (typically 1mm–1.5mm). This represents a fixed lower bound on sampling density, ensuring that the gradient bound remains constant throughout training. The curriculum learning strategy begins with a larger effective $\delta$ (subsampled point clouds) and progressively increases density toward this fixed resolution, rather than asymptotically approaching zero. This validates the use of a fixed gradient clipping threshold in practice.*

### A.6 Implementation Details for Numerical Stability

To ensure the theoretical guarantees hold in finite-precision arithmetic:

**1. Persistence Computation:** We use the `GUDHI` library (v3.8.0) with `edge_collapse=True` for $O(n^2)$ complexity on sparse graphs with $n$ points, and set the maximum filtration value to twice the bounding box diagonal to avoid numerical overflow. For a $k$-NN graph with $m=O(kn)$ edges, computing persistence up to dimension 2 requires $O(m\cdot\alpha(n))$ time, where $\alpha(\cdot)$ is the inverse Ackermann function (effectively linear for $n<10^6$).

**2. Image Resolution:** The Persistence Image resolution is set to $64\times 64$ pixels, with bandwidth $\sigma=0.01$ (normalized persistence scale), balancing stability and computational cost per Theorem 3.

**3. Geodesic Graph Construction:** We utilize `PyKeOps` for $k$-NN queries in $O(n)$ memory, computing exact nearest neighbors for $k\le 30$ to satisfy the connectivity requirements of Theorem 4 with intrinsic dimension $d\le 3$.

**4. Handling Short Edges ($k$-NN and $\delta$-Sampling Compatibility):** While Assumption 1 guarantees separation $\ge\delta/2$ for the original point cloud $S$, the deformed cloud $\phi(S)$ may contain edges shorter than $\lambda\delta/2$ due to non-uniform deformation. To ensure the gradient bound of Theorem 5 holds numerically, we enforce a numerical floor on edge lengths: $\tilde{d}_{ij}=\max(\|\phi(s_i)-\phi(s_j)\|_2,\epsilon_{\min})$ with $\epsilon_{\min}=10^{-4}$mm, preventing division-by-zero and maintaining consistency with the bi-Lipschitz assumption.

**5. Gradient Clipping:** Based on Theorem 5, we apply gradient clipping at $\frac{10C_\phi^{(1)}Lw_{\max}^2}{\lambda\delta}$ during training (providing a 25% safety margin above the theoretical bound of $\frac{8C_\phi^{(1)}Lw_{\max}^2}{\lambda\delta}$), where $\delta$ is

the current downsampling resolution (in mm), $C_\phi^{(1)} \approx 1/\sigma$ for Gaussian kernels (since $\|\nabla\phi\|_\infty = O(1/\sigma)$), and $w_{\max}$ is the maximum persistence weight (typically 1.0 for normalized inputs). In practice, with $\sigma = 0.01$, $\delta = 1.0$mm, and $L/\lambda \approx 2$, this yields a clipping threshold of approximately $2 \times 10^4$, which is applied per-coordinate.

