# OpenReview forum: "TopoPointPWC: Manifold Topology-Aware Point Cloud Registration via Persistent Homology"
_ICLR.cc/2026/Workshop/GRaM — ICLR 2026 Workshop GRaM Poster_

### Official Review · Reviewer_4uG3 · 2026-02-09
**A topology aware pipeline for non-rigid point cloud registration**

**Rating:** 4
**Confidence:** 3

**Review:**

This paper proposes a topology-aware pipeline for non-rigid point cloud registration. The method first constructs a manifold-based kNN graph by computing shortest-path distances on the Euclidean kNN graph. Then, they extract features from this manifold kNN graph and embed them as persistence images. They inject topological constraints via channel-wise attention to guide via topology.

My primary concern is the lack of experimental validation of the final model. While there is empirical evidence that the manifold kNN graph has more desirable properties (such as not allowing shortcuts) than the Euclidean kNN one, this is not particularly surprising. It is unclear how much these properties matter in downstream use. I would expect to see an evaluation of the registration of the full method introduced in the paper, as without this, the impact is difficult to assess.

**Pmlr Suitability:**

NA

---

### Official Review · Reviewer_BpSN · 2026-02-23
**Manifold-respecting anatomical point cloud registration is promising but preliminary**

**Rating:** 4
**Confidence:** 3

**Review:**

**Summary**:
This paper proposes TopoPointPWC, a manifold-based point cloud registration method for anatomical structures. The approach is well-motivated since operating on learned manifolds rather than in ambient Euclidean space should better respect the topology of curved anatomical surfaces. The method integrates this idea into a hierarchical PWC-style registration pipeline.

**Strengths**:
* The manifold formulation is intuitively appropriate for anatomical data and aligns well with the underlying geometry.
* The overall architecture is sensible and targets a meaningful application area.

**Weaknesses**:
* The empirical evaluation is not extensive enough to support the claims. There is no ablation of the loss terms or curriculum learning strategy, so it is unclear what drives performance.
* The paper is sometimes hard to follow. Several acronyms are introduced before being defined (e.g., FPS and TRE in lines 93-94), and parts of the technical setup (e.g., the distance metric in Theorem 1 or the explanation of how the dense field $\phi$ is extracted in lines 137-138) were unclear.
* The connection between the theoretical discussion and empirical gains could be better articulated
* Robustness/scalability are not deeply explored

**Overall assessment**:
The manifold-based perspective is well-motivated and promising for anatomical registration, but clearer writing and stronger empirical validation would make the contribution more convincing.

**Minor comments**:
* Typo on line 30: "rendering registration a highly ill-posed problem" appears twice
* Citations need to be properly formatted to be parenthetical when not used in-text

**Pmlr Suitability:**

NA

---

### Meta-Review · Area_Chair_5gsf · 2026-02-25

**Decision:**

Accept

**Metareview:**

While the reviews are critical and identified important points for improvement esp. in the presentation, the idea is original and creative, and worth including at GRaM in the non-archival tiny paper track. In my decision I lowered my weighting on the experimental validation, as for these type of registration tasks this is challenging, and perhaps more important for a full paper submission. I do encourage the authors to improve the clarify of writing for a possible cam ready version.

**Relevance To Proceedings:**

Tiny paper — does not apply

**Relevance To Workshop:**

Yes — suitable for GRaM

---

### Decision · Program_Chairs · 2026-03-02

Accept (Poster)